# Improving Genomic Prediction Accuracy in the Chinese Holstein Population by Combining with the Nordic Holstein Reference Population

**DOI:** 10.3390/ani13040636

**Published:** 2023-02-11

**Authors:** Zipeng Zhang, Shaolei Shi, Qin Zhang, Gert P. Aamand, Mogens S. Lund, Guosheng Su, Xiangdong Ding

**Affiliations:** 1National Engineering Laboratory for Animal Breeding, College of Animal Science and Technology, China Agricultural University, Beijing 100193, China; 2Shandong Provincial Key Laboratory of Animal Biotechnology and Disease Control and Prevention, Shandong Agricultural University, Taian 271018, China; 3Nordic Cattle Genetic Evaluation, 8200 Aarhus, Denmark; 4Center for Quantitative Genetics and Genomics, Aarhus University, 8830 Tjele, Denmark

**Keywords:** genomic prediction, joint reference population, genotype by environment interaction, multi-trait GBLUP

## Abstract

**Simple Summary:**

The size of the reference population is critical to the accuracy of genomic prediction. In addition, joining the reference populations from different breeding organizations is a convenient and effective method by which to enlarge reference populations. By adding the Nordic Holstein reference population to the Chinese Holstein reference population, we found that the accuracy of genomic prediction in the Chinese Holstein population was improved substantially for the traits with high or moderate genetic correlation between the two populations; however, the low-genetic-correlation traits did not improve. These findings are important for the purposes of multi-country joint genomic evaluation.

**Abstract:**

The size of the reference population is critical in order to improve the accuracy of genomic prediction. Indeed, improving genomic prediction accuracy by combining multinational reference populations has proven to be effective. In this study, we investigated the improvement of genomic prediction accuracy in seven complex traits (i.e., milk yield; fat yield; protein yield; somatic cell count; body conformation; feet and legs; and mammary system conformation) by combining the Chinese and Nordic Holstein reference populations. The estimated genetic correlations between the Chinese and Nordic Holstein populations are high with respect to protein yield, fat yield, and milk yield—whereby these correlations range from 0.621 to 0.720—and are moderate with respect to somatic cell count (0.449), but low for the three conformation traits (which range from 0.144 to 0.236). When utilizing the joint reference data and a two-trait GBLUP model, the genomic prediction accuracy in the Chinese Holsteins improves considerably with respect to the traits with moderate-to-high genetic correlations, whereas the improvement in Nordic Holsteins is small. When compared with the single population analysis, using the joint reference population for genomic prediction in younger animals, results in a 2.3 to 8.1 percent improvement in accuracy. Meanwhile, 10 replications of five-fold cross-validation were also implemented in order to evaluate the performance of joint genomic prediction, thereby resulting in a 1.6 to 5.2 percent increase in accuracy. With respect to joint genomic prediction, the bias was found to be quite low. However, for traits with low genetic correlations, the joint reference data do not improve the prediction accuracy substantially for either population.

## 1. Introduction

Genomic selection (GS) [1] is a new landmark method of genetic evaluation that follows the BLUP method [2]. It has already become a tool for routine genetic evaluation in regard to dairy cattle breeding. The principle of GS is that at least one genetic marker in the whole genome is in linkage disequilibrium with the quantitative trait loci of the trait of interest [1]. These genetic markers are used to estimate the effect of each QTL and then calculate the genomic estimated breeding value for the target trait. In addition, genetic markers can capture the relationship of individuals who cannot be traced to a common ancestor in the pedigree. When compared with the traditional BLUP method, genomic selection substantially improves the prediction accuracy in selection candidates and shortens the generation interval [3,4]. In the United States, seven years after the introduction of genomic selection in dairy cattle, the annual rates of genetic gain increased from 50% to 100% for yield traits, and from three-fold to four-fold for lowly heritable traits—including female fertility, herd life, and somatic cell count [5].

The reference population size is one of the most important factors for the purposes of improving genomic prediction accuracy [6,7]. A cost-effective approach is to use genotype imputation as a strategy in order to reduce cost or to increase/maintain the prediction accuracy for selection candidates and to render genomic selection economically feasible. Another cost-effective approach to increase the reference population size is to combine with other reference populations. Furthermore, it has been reported that combining multiple reference populations can increase genomic prediction accuracy [8,9,10]. Indeed, with the help of frozen semen technology, the genetic material of Holstein dairy cattle has been delivered globally. As such, this fact has provided a genetic basis for the joint genomic evaluation of dairy cattle.

When combining multiple populations in order to enlarge the reference population, the consistency of linkage disequilibrium (LD) among the populations requires consideration. If high LD consistency exists among the respective populations, then combining the different populations is expected to improve the genomic prediction accuracy considerably. With respect to this, the EuroGeomics Consortium [9] and the North American Consortium [11] combined multiple, genetically related populations into a large reference population, thereby resulting in a significant improvement with respect to genomic prediction accuracy. In addition, the combination of the Holstein populations from different countries for the purposes of cross-country joint assessment resulted in good outcomes for the countries who possessed relatively small reference populations. For instance, Zhou et al. combined both Chinese and Nordic Holstein populations, which possessed highly consistent LD (0.97), as a joint reference population. As such, they found a substantial improvement in the genomic prediction accuracy of milk production traits in Chinese populations [12]. In addition, Li et al. found that prediction reliabilities for the Brazilian Holstein population could be greatly increased by including the data of Nordic and French Holstein populations [8]. Meanwhile, certain studies have, instead, shown that multi-population or multi-breed genomic selection, via the combination of distantly related populations, could not improve prediction accuracy and even, in certain circumstances, reduced it [13,14,15].

Genotype-by-environment interaction (G × E) is also a key factor that influences the accuracy of multi-population genomic selection. This is due to the fact that many economically important traits are influenced by a combination of genetics, environment, and G × E. Moreover, different climates, housing conditions, nutritional levels, disease pressure, and feeding densities may also cause G × E. Furthermore, when phenotypes of different reference populations are measured in different environments, ignoring the influence of G × E decreases the gain in genomic prediction accuracy when combining reference populations. As such, the biases may increase [16] due to the fact that the LD differences can lead to an increased bias in the estimation of SNP effects.

It must be noted that genomic selection has been carried out in Chinese Holstein breeding since 2012. While the reference population mainly comprised cows [17], the number of progeny-tested bulls in the reference population has been increasing in recent years. It is well-known that in a cattle population, a breeding bull possesses a large number of daughters; thus, the phenotypic information of a bull is of greater importance than that of a cow. The Chinese Holsteins were originally imported from Europe and North America, whereby frozen semen and embryos were imported from abroad in recent decades. Zhou et al. reported very high LD concordance with respect to the adjacent markers between the Chinese and Nordic Holstein populations, as based on 54 K marker data and with a correlation of 0.97 [12]. Therefore, after including the bull reference population of the Nordic Holstein, which consisted of progeny-tested bulls, the genomic prediction accuracy for the Chinese population is expected to improve considerably.

The objective of this study, therefore, was to investigate the improvement of genomic prediction in the Chinese Holstein population, with a joint reference population that consists of Chinese and Nordic Holsteins. We also tested the genomic prediction accuracy using cross-population prediction, i.e., the Chinese Holstein reference population was used for the prediction of Nordic Holsteins, and the Nordic Holstein reference population was used for prediction of Chinese Holsteins.

## 2. Materials and Methods

### 2.1. Phenotypes

The dataset utilized in this study was derived from 9206 Chinese Holsteins with genotypes born between 1984 and 2018 and 7084 Holsteins with genotypes born between 1977 and 2012 in certain Nordic countries (including Denmark, Finland, and Sweden). The Chinese Holstein population comprises 5333 bulls and 3873 cows, while the Nordic Holstein population consists of all bulls. Seven traits—milk yield (MY), fat yield (FY), protein yield (PY), somatic cell score (SCS) (with respect to the Chinese Holsteins) or somatic cell count (SCC) (regarding the Nordic Holsteins), body conformation (CONF), feet and legs (FL), and mammary system conformation (MS)—were analyzed. In the Chinese population, 6000 individuals with records were available for MY, FY, and PY, as well as 7000 individuals that possessed records for the other traits. In regard to the Nordic population, the number of individuals with phenotypes for MY, FY, and PY was the same as in the Chinese population. However, the 6495 Holsteins did possess phenotypes for CONF, FL, and MS, and 6347 Holsteins possessed phenotypes for SCC. In the two populations, the numbers of individuals for the three milk-production traits were equal. In addition, the numbers of individuals for the three types of traits were also the same. The de-regressed proof (DRP) was a standardized DRP with respect to the Nordic population, but the original DRP was revealed as a deviation from the base population in the Chinese population. The summary statistics of the DRP for each trait are listed in Table 1. Among them, in regard to the Nordic Holstein population, the averaged reliabilities of the DRP for both milk-production traits and the SCC were found to be higher than or equal to 0.914; further, the reliabilities of the DRP on type traits ranged from 0.710 to 0.844. In addition, the averaged reliability of the DRP in the Chinese Holstein population was found to be lower. Moreover, the milk-production traits and SCS possessed a reliability of 0.502 to 0.660 on average, and the reliabilities of DRP on all type traits ranged from 0.401 to 0.409.

### 2.2. Genotypes and Quality Control

All the individuals in both populations were genotyped with Illumina BovineSNP50 BeadChip (Illumina Inc., San Diego, CA, USA). For each population, the SNPs with a minor allele frequency (MAF) of less than 0.01 were deleted. Further, SNPs with a call rate of less than 0.90 were removed. After genotype quality control was conducted, the individuals with a call rate of less than 0.90 were excluded.. For the remaining individuals, the SNPs with missing genotypes were imputed via using Beagle 3.31 software [18]. The SNPs with an imputation accuracy (R square) less than 0.95 were also removed. As such, after genotype quality control was performed, a total of 43,613 autosomal SNPs, but no animals with a call rate of less than 0.90.

### 2.3. Phenotypes and Quality Control

We analyzed the following three scenarios:

(1) A single-trait GBLUP model was used in order to predict direct genomic breeding values (DGV) while using thereference data of the same population;

(2) A single-trait GBLUP model was used to predict the DGV while using the reference data of a population in order to predict another population (i.e., using the Nordic reference population in order to predict the Chinese validation population);

(3) A two-trait GBLUP model was used for the purposes of genomic prediction while using the joint Chinese–Nordic population, in which the same trait in the different populations was treated as different, but as genetically correlated traits.

The variance components and DGV were estimated using DMU [19]. Furthermore, the algorithm used to estimate variance components was determined by the average information that restricted the maximum likelihood (REML) [20,21].

#### 2.3.1. Single-Trait GBLUP

The single-trait GBLUP model was defined as
y=1μ+Zg+e
where y is the vector of $\begin{array}{c} n \end{array}$ pseudo-phenotypes (i.e., the de-regressed proof (DRP)); 1 is a vector of 1 s; μ is the overall mean; g is the vector of additive genetic values with respect to the individuals with a genotype; Z is the design matrix linking the phenotype to the genetic values; and e is the vector of the random residuals. The assumptions of the random effects are:$\;g\sim N\left(0,G\sigma_{g}^{2}\right)$ and $e\sim N\left(0,D\sigma_{e}^{2}\right)$. Furthermore, $\sigma_{g}^{2}$ and $\sigma_{e}^{2}$ are the addictive genetic variance and residual variance, respectively. In addition, G represents the genomic relationship matrix, which was calculated with method 1, as was described by VanRaden [22]. Lastly, D is the diagonal matrix of the weights for the residuals [23], where the diagonal elements were calculated as $d_{ii}=\frac{1-r_{i}^{2}}{r_{i}^{2}}$, and $r_{i}^{2}$ is the reliability of the DRP with respect to the individual $\begin{array}{c} i \end{array}$.

#### 2.3.2. Two-Trait GBLUP

Two-trait GBLUP model is:$$\begin{array}{c} \begin{array}{c} \left[\begin{array}{c} y_{1}\\ y_{2} \end{array}\right] \end{array} \end{array}=\left[\begin{array}{cc} 1_{n_{1}} & 0\\ 0 & 1_{n_{2}} \end{array}\right]\left[\begin{array}{c} \mu_{1}\\ \mu_{2} \end{array}\right]+\left[\begin{array}{cc} Z_{1} & 0\\ 0 & Z_{2} \end{array}\right]\left[\begin{array}{c} g_{1}\\ g_{2} \end{array}\right]+\left[\begin{array}{c} e_{1}\\ e_{2} \end{array}\right]$$
where $y_{1}$ and $y_{2}$ are the vectors of DRP for trait 1 and trait 2, thus, corresponding to the pseudo observations in the Chinese Holstein and Nordic Holstein populations, respectively. Furthermore, $\mu_{1}$ and $\mu_{2}$ are the overall means of the two traits, and $g_{1}$ and $g_{2}$ are the vectors of the additive genetic values of the two traits. Moreover, it was assumed that $\left[\begin{array}{c} g_{1}\\ g_{2} \end{array}\right]\sim N\left(0,G_{0}⊗G\right)$, whereby $\;G_{0}$ is the genetic variance and the covariance matrix for the two traits $G_{0}=\left[\begin{array}{cc} \sigma_{g_{1}}^{2} & \sigma_{g_{1}g_{2}}\\ \sigma_{g_{1}g_{2}} & \sigma_{g_{2}}^{2} \end{array}\right]$. Then, G is the genomic relationship matrix of all genotyped animals, and $\sigma_{g_{1}}^{2}$ and $\sigma_{g_{2}}^{2}$ are the additive genetic variance with respect to the two populations. Further, $\sigma_{g_{1}g_{2}}$ is the genetic variance-covariance matrix between the two populations. Next, $e_{1}$ and $e_{2}$ are the vectors of the random residuals of the two populations, which were assumed to be independent of each other with respect to $e_{1}\sim N\left(0,D_{1}\sigma_{e}^{2}\right)$ and $e_{2}\sim N\left(0,D_{2}\sigma_{e}^{2}\right)$. The construction of G, $D_{1}$, and $D_{2}$ were the same as ***G*** and ***D***, which were described in the above single-trait GBLUP model.

#### 2.3.3. Genomic Prediction Accuracy

The genomic prediction accuracy and prediction unbiasedness were obtained through two ways. The first was in terms of predicting the younger individuals that were using the older animals. All the individuals from the Chinese and Nordic Holstein populations were divided into reference and validation population sets, with a ratio around 4:1 in population size and according to the cut-off date as shown in Table 1. The 1/5 younger individuals were also used as the validation population, thereby called the Chinese validation population and the Nordic validation population. The remaining 4/5 individuals of each population were used as the reference population, thus, called the Chinese reference population and the Nordic reference population, respectively. Alternatively, a five-fold cross-validation was also implemented. The prediction accuracy and prediction unbiasedness were obtained through five-fold cross-validation. For each trait, the individuals in both populations were randomly split into five groups, in which the numbers of individuals from the Chinese population and the Nordic population were nearly the same among the five groups. In each round of cross-validation, one group was taken as the validation population, the other four groups were used as the reference population. Similarly, either with respect to the Chinese population or the Nordic population, the five-fold cross-validation of a single population was carried out via using their own five groups. Furthermore, the five-fold cross-validation was replicated a total of 10 times.

The prediction accuracy of the GBLUP was calculated as
$$\begin{array}{c} acc=\frac{cor\left(DRP,DGV\right)}{\bar{r}} \end{array},$$
where cor(DRP,DGV) is the correlation between the *DRP* and the *DGV* for the animals in the validation population. In addition, $\bar{r}$ is the averaged accuracy of the *DRP* in the validation population.

Next, the prediction unbiasedness of GBLUP was calculated as:(1)$$b=\frac{cov\left(DRP,DGV\right)}{var\left(DGV\right)}$$
where *cov(DRP,DGV)* is the covariance between the *DRP* and the *DGV* for the animals in the validation population. Further, the *var (DGV)* is the variance of the *DGV* in the validation population.

## 3. Results

### 3.1. Genetic Correlation of Traits in Two Holstein Populations

Table 2 shows the genetic correlations between the Chinese and Nordic Holstein populations, which were estimated with a two-trait GBLUP model. Regarding the three milk-production traits (i.e., MY, FY, and PY), the genetic correlations are high and range between 0.621 to 0.720. Moreover, the SCS possess a genetic correlation of 0.449. Regarding the type traits (i.e., CONF, FL, and MS), the genetic correlations from the Chinese and Nordic Holstein populations are low (i.e., less than or equal to 0.236). Moreover, the genetic correlation between the same trait in the two populations indicates that G×E exists [24]. 

Regarding the individuals with milk-production trait records, the average genomic relationships withinthe Chinese and Nordic Holstein populations are 0.030 (±0.043) and 0.033 (±0.038), respectively. The average genomic relationship between the two populations is 0.022 (±0.022), which is lower than the average genomic relationships within the populations. With respect to the individuals with type trait records, the average genomic relationships withinthe Chinese and Nordic Holstein populations are 0.029 (±0.039) and 0.033 (±0.037), respectively. In addition, the average genomic relationship between the two populations is the same as that found in the milk-production traits. Regarding the individuals with SCC/SCS records, the average genomic relationship of the Chinese and Nordic Holstein populations is 0.029 (±0.041) and 0.033 (±0.037), respectively. However, the average genomic relationship between the two populations is 0.022 (±0.021).

### 3.2. Comparison between Single Population and Joint Population Prediction Accuracy

Table 3 shows the prediction accuracy and unbiasedness of the younger validation individuals with respect to both the Chinese and Nordic Holstein populations, which was achieved using the own- and joint-reference populations. In regard to the milk-production traits (i.e., MY, FY, and PY) and the SCS in the Chinese Holstein population, the genomic prediction accuracies using the single Chinese reference population (i.e., as per the single-trait model) are 0.402, 0.429, 0.394, and 0.253, respectively. Indeed, the genomic prediction accuracy via using the joint reference population (i.e., the two-trait model) reaches 0.464, 0.510, 0.433, and 0.276, respectively. Moreover, the joint reference population gains 2.3 to 8.1 percentage points over the single reference population with respect to the accuracy of genomic selection for the Chinese Holstein population. However, in regard to the Nordic Holstein population, the prediction accuracy is improved very little when compared with the own reference population. Further, the MY trait gains the greatest improvement in terms of prediction accuracy, albeit only from 0.683 to 0.703. With respect to the type traits, the prediction accuracy, with a negligible change of −0.4 to 0.2 percentage points when compared to their single own reference groups, is not improved in either of the Chinese orNordic Holstein populations.

For the Chinese Holstein population, the unbiasedness of genomic prediction ofthe milk-production traits and the SCS is significantly improved when using the joint reference population. In particular, the unbiasedness of genomic prediction regarding SCS dramatically improves from 0.726 to 0.927. However, for the type traits, the joint reference population leads to a bigger bias. For example, the the genomic prediction unbiasedness of MS is 0.989 when using the single Chinese reference population, but decreases to 0.946 when using the joint reference population. However, the Nordic Holstein population does not benefit from the joint reference population with respect to the prediction unbiasedness, as is the case in regards to accuracy.

Table 4 further details the prediction accuracy and unbiasedness, from the single and joint reference populations in the 10 replicates of the five-fold cross-validation, for both the Chinese and Nordic Holstein populations. When compared with the single population, the impact of the joint reference data regarding the genomic prediction from the cross-validation is consistent with that of the younger validation individuals. Nonetheless, the improvement in the prediction accuracy of the milk-production traits and the SCS for the Chinese population is found to be lower, ranging from 1.6 to 5.2 percentage points, than the prediction accuracy of the younger validation individuals. Indeed, the changes in unbiasedness for all traits are small, and the genomic prediction unbiasedness obtained from the cross-validation is generally found to be better than that from the younger validation individuals, which is closer to 1.0.

### 3.3. The Genomic Prediction Accuracy of Cross Population

In addition, we analyzed the genomic prediction with respect to the cross population, as demonstrated in Table 5 and Table 6. Regarding the younger validation individuals, the prediction accuracy from the cross population (e.g., when predicting with respect to the Chinese Holstein population using the Nordic reference population) is found to be worse than the prediction that was obtained using their own, respective, reference population. More specifically, there is a large decrease in the prediction accuracy of the type traits. For example, the prediction accuracies of FL, for the validation population of the Chinese and Nordic Holsteins, are found to be 0.493 and 0.589 (Table 3) when compared with their own reference populations, respectively. However, the prediction accuracies of the FL, when estimated from the cross population, are only 0.088 and 0.212 (Table 5) for the Chinese and Nordic Holstein populations, respectively. This, therefore, implies the importance of the own reference population. Similar results are also found in regard to the other traits, but the decrease in accuracy when using the foreign reference population varies. For example, the decrease in genomic prediction accuracy for milk-production traits is found to be relatively small. However, the prediction accuracy of the MY trait in the Chinese validation population predicted by the Chinese reference population is 0.402, while the accuracy of prediction when using the Nordic reference population is 0.354. Similarly, as shown in Table 6, the performance of the cross population in regard to the prediction accuracy obtained in the 10 replicates of five-fold cross-validation is found to be worse, even when compared with the own or joint reference population.

## 4. Discussion

In this study, we investigated whether enlarging the Holstein reference population, by combining the Chinese and Nordic Holstein reference populations, could improve the accuracy of genomic selection in the two Holstein populations. The results show that the genomic prediction of the milk-production traits (i.e., MY, FY, and PY) and SCC in the Chinese population is improved substantially using the joint reference population, albeit this is not the case for the type traits. However, the Nordic Holstein population does not gain much improvement in terms of the genomic prediction of all the traits. Although the reference population size of the Chinese and Nordic Holstein is nearly equal (Table 1), the Nordic reference population provides much more information. This is due to the fact that the Nordic reference population consists of progeny-tested bulls that possess a higher DRP reliability with respect to the concerned traits. Therefore, the joint reference population is found to be more helpful regarding the Chinese Holstein population. Regarding the aforementioned, Lund et al. also demonstrate that the benefit that is obtained from the joint reference population is influenced by the DRP reliability of the foreign reference population [9]. Moreover, in the joint genomic prediction of the Holstein and Jersey populations (which was conducted in Lund et al.’s study), it was also indicated that the Jersey population, which possessed a smaller reference population, gained more of an improvement in terms of genomic prediction, while the Holstein population acquired only a slight improvement due to its larger reference population [15].

On the other hand, the accuracies of the genomic selection for all traits in the Nordic Holstein population are higher than in those in the Chinese Holstein population, when their own reference population (i.e., the progeny-tested bulls) is used only to further demonstrate the composition of the reference population, which is essential for the purposes of genomic selection. Nevertheless, it is feasible to add cows with genotypic information in order to enlarge the reference population for the purposes of improving the genomic prediction accuracy when the number of bulls is insufficient [25]. Indeed, Ding et al. [17] investigated if it is plausible to use cows as the Holstein reference population. When 3084 cows were used as the reference population, the correlation between the genomic EBV and the conventional EBV, regarding the validation population of the five milk-production traits, went as high as from 0.594 to 0.760. Moreover, Mc Hugh et al. also showed that the genomic information that is obtained from cows could improve the genomic prediction accuracy [26]. In addition, cows can be genotyped using low-density chips and then imputed with high-density chips in order to achieve a low-cost increase in the cow reference population size [27].

Indeed, the size of the reference population is a critical factor regarding genome prediction [6,7]. Combining populations is a simple and practical cost-efficient strategy in order to improve genomic prediction accuracy. Moreover, Li et al. found that adding Nordic and French bull data in order to enlarge the Brazilian reference population could improve the genomic prediction reliability of the three milk-production traits by 0.030 to 0.055 [8]. Similar results were obtained by adding 870 foreign Brown Swiss cows to the US Brown Swiss reference group. Further, the genomic prediction accuracy was improved by 3.2%, on average, with a single-trait model, and by 4.6% with a two-trait model [28]. In our study, only a two-trait model was used to estimate the genomic breeding values. This was decided upon due to the fact that the single-trait model could not be carried out due to the different scales with respect to the DRP in the Chinese and Nordic populations. Moreover, the two-trait model could also account for G × E regarding the two populations [29]. Meanwhile, our results also indicate that the joint reference population is helpful with respect to improving the genomic prediction accuracy of the milk-production traits and the SCS/SCC, while no such improvements are gained regarding the type traits. The reason for this could be that the genetic correlations of the three type traits (i.e., CONF, FL, and MS) are much lower than those of the milk-production traits and the SCS/SCC (Table 2). Indeed, the high genetic correlations mean that the foreign population could provide more information and increase the efficiency of using the joint reference population. This point was confirmed in the study of dairy cattle in the EuroGenomics project that was reported by Lund et al. A 12–19% increase, in the project, with respect to the genomic prediction accuracy was gained on udder depth, as this trait possessed the highest genetic correlation (0.98) among the collaborating countries. In contrast, the genetic correlation for the SCS was relatively low (0.88), thereby resulting in an increase in accuracy of 8–15% [9]. In our study, the genetic correlations of the type traits between the Chinese and Nordic Holstein populations are quite low (0.144–0.236), which may be due to the different trait definitions or measurement methods for these type traits in the two populations. Consequently, combining the two reference populations does not provide more information than using a single population. Thus, no improvements in genomic prediction accuracy are obtained.

Although the Chinese Holstein cows mainly originated from Europe and North America, the differences in climate, feeding environment, selection criterion, etc., between China and the Nordic countries resulted in different performances with respect to the two Holstein populations. This could be explained by the existence of G × E in these two populations. Robertson proposed that a genetic correlation less than 0.80 between the same trait in two different environments indicates the existence of G × E a [24]. According to this proposal, all of the traits in this study showed G × E in the Chinese and Nordic populations, e.g., milk-production traits possess a high genetic correlation (0.621–0.720), SCC/SCS are moderately genetically correlated (0.449), and the type traits are lowly genetically correlated (0.144–0.236). In the dairy cattle populations in the Eurogenomics project, the G × E could be ignored in most traits, e.g., udder depth and SCC, as their genetic correlations were greater than 0.8. Furthermore, these traits could, thus, be treated as one trait, i.e., a single-trait model using the joint reference population could be a good approach [9]. However, in the scenario with G × E, it is not reasonable to implement a single-trait model in terms of genomic selection, particularly with respect to the traits with low genetic correlations. In such situations, the single-trait model may generate large bias using a combined reference population, while the two-trait model could be of better use due to its accounting for G × E [30]. When treating the same trait of two populations as different traits, it enables one to capture G × E as a covariance between the populations, which, in turn, allows one to account for G × E in the model [31]. In addition, the two-trait model is more flexiblewhen the scales used to measure phenotypes for the same trait are different in the different populations. For example, the DRP is a standardized DRP in the Nordic population, but the original DRP serves as a deviation from the base population in the Chinese population in our study; as such, the single-trait model would, thus, be implausible.

The reliability of the DRP for each trait is found to be much higher in the Nordic population than in the Chinese population. As such, we, therefore, analyzed whether the Nordic reference population could yield a higher genomic prediction accuracy for the Chinese Holstein population than the Chinese reference population could. The results of the cross-population prediction (Table 5) show that the accuracies are worse when compared with using the own reference population. Likewise, the Chinese reference population does not yield a reasonable prediction accuracy with respect to the Nordic population. However, it is consistent with the report by Ma et al. regarding the prediction of the Chinese Holstein population when using French and USA reference populations [32]. Our results suggest that the foreign reference population, in general, cannot lead to the higher accuracy of prediction that is found in the Chinese reference population. Having said this, however, the LD is found to be similar between the two cattle populations [12], while a joint reference population including the foreign reference population to the Chinese reference populations is found to be helpful.

The genomic Chinese performance index (GCPI) that is currently used in the genomic breeding of the Chinese Holstein cows consists of the seven traits that are investigated in this study. The improvement in the prediction accuracy, regarding the production traits and the SCS traits, is achieved by combining the Nordic population with the Chinese population, which is, thus, found to be helpful for the purposes of Chinese Holstein breeding. Moreover, the longevity traits, reproductive traits, feed conversion traits, etc., will also be gradually added to the GCPI. The information regarding the reference population from the Nordic countries and other countries/organizations would be also helpful in order to improve the genomic prediction efficiency, as indicated in this study. Therefore, it is necessary to carry out this joint genomic selection for China and for other countries.

## 5. Conclusions

Genomic prediction when using a foreign reference population may not obtain high accuracy. However, the joint reference population can improve the prediction accuracy and prediction unbiasedness. In particular, this is true for the traits that possess moderate-to-high genetic correlations between the populations. As for traits that possess low genetic correlations between the populations, a joint reference population may not improve the prediction accuracy and prediction unbiasedness. However, the difference in prediction accuracy when using own, foreign, and joint reference populations is also dependent on the composition of the reference populations.

## Figures and Tables

**Table 1 animals-13-00636-t001:** Descriptive statistics of DRP ^a^ regarding the Chinese and Nordic Holstein populations.

		Milk Yield (kg)	Fat Yield (kg)	Protein Yield (kg)	Body Conformation	Feet and Legs	Mammary System Conformation	Somatic Cell Score (10^3^/mL)
Number of Records	Nordic	6000	6000	6000	6495	6495	6495	6347
	Chinese	6000	6000	6000	7000	7000	7000	7000
Cut-off date ^b^	Nordic	December 2006			January 2017			April 2007
	Chinese	July 2013			September 2013			June 2013
DRP reliability	Nordic	0.945	0.945	0.945	0.844	0.710	0.803	0.914
	Chinese	0.656	0.660	0.658	0.401	0.409	0.403	0.502
Max	Nordic	132.8	130.3	146.8	159.2	201.4	160.3	150.4
	Chinese	4907.8	192.5	130.0	51.5	48.5	48.4	389.5
Min	Nordic	45.0	36.7	7.7	56.8	−60.0	−7.3	51.1
	Chinese	−5390.9	−185.9	−182.0	−43.8	−37.2	−59.4	215.3
Mean	Nordic	93.6	93.4	90.7	100.0	94.8	91.1	94.7
	Chinese	261.3	3.6	7.7	−1.8	−1.3	−1.5	300.1
S.D	Nordic	12.7	12.3	13.8	12.4	15.4	14.9	11.7
	Chinese	1237.1	47.0	39.5	10.1	9.7	10.9	18.7

^a^ The DRP is the standardized DRP in the Nordic population, but the original DRP is understood as the deviation from the base population with respect to the Chinese population. ^b^ Cut-off date in order to divide the whole data into reference and validation sets.

**Table 2 animals-13-00636-t002:** Genetic correlation of traits between Chinese and Nordic Holstein populations.

Traits	Variance		Covariance	Correlation
	Chinese	Nordic		
Milk yield	470,936.770	102.627	5005.297	0.720 (0.030)
Fat yield	634.546	175.126	94.554	0.715 (0.031)
Protein yield	422.192	124.551	95.1678	0.621 (0.035)
Body conformation	29.540	12.190	102.790	0.221 (0.043)
Feet and legs	40.368	9.601	109.742	0.144 (0.044)
Mammary system conformation	31.657	14.707	123.003	0.236 (0.044)
Somatic cell score	58.536	34.502	101.069	0.449 (0.046)

**Table 3 animals-13-00636-t003:** Prediction accuracy and unbiasedness regarding the younger individuals with respect to the seven traits in the Chinese and Nordic Holstein populations, as predicted by the single/joint reference populations.

Reference	Validation		Milk Yield	Fat Yield	Protein Yield	Body Conformation	Feet and Legs	Mammary System Conformation	Somatic Cell Score
Single	Nordic	Accuracy	0.683	0.681	0.654	0.752	0.589	0.703	0.682
		Unbiasedness	0.823	0.807	0.722	0.993	0.898	0.974	0.987
	Chinese	Accuracy	0.402	0.429	0.394	0.439	0.493	0.507	0.253
		Unbiasedness	0.937	0.948	0.860	0.839	0.800	0.989	0.726
Joint	Nordic	Accuracy	0.703	0.696	0.665	0.753	0.591	0.704	0.687
		Unbiasedness	0.842	0.815	0.731	1.002	0.89	0.973	0.983
	Chinese	Accuracy	0.464	0.510	0.433	0.441	0.489	0.507	0.276
		Unbiasedness	0.976	0.983	0.898	0.82	0.773	0.946	0.927

**Table 4 animals-13-00636-t004:** Prediction accuracy and unbiasedness regarding the seven traits in the Chinese and Nordic Holstein populations, as predicted by the single/joint reference populations in the 10 replicates of the five-fold cross-validation.

Reference	Validation		Milk Yield	Fat Yield	Protein Yield	Body Conformation	Feet and Legs	Mammary System Conformation	Somatic Cell Score
Single	Nordic	Accuracy	0.860 (±0.007)	0.854 (±0.009)	0.897 (±0.007)	0.783 (±0.013)	0.707 (±0.024)	0.853 (±0.016)	0.763 (±0.015)
		Unbiasedness	1.006 (±0.021)	1.004 (±0.018)	1.009 (±0.016)	0.989 (0.033)	1.002 (±0.046)	1.016 (±0.027)	0.982 (±0.030)
	Chinese	Accuracy	0.502 (±0.024)	0.475 (±0.026)	0.507 (±0.024)	0.590 (±0.033)	0.680 (±0.035)	0.590 (±0.037)	0.413 (±0.031)
		Unbiasedness	1.086 (±0.080)	1.072 (±0.079)	0.897 (±0.007)	0.943 (±0.069)	0.904 (±0.054)	0.989 (±0.085)	1.212 (±0.111)
Joint	Nordic	Accuracy	0.864 (±0.007)	0.860 (±0.023)	0.902 (±0.019)	0.783 (±0.013)	0.708 (±0.024)	0.853 (±0.016)	0.766 (±0.015)
		Unbiasedness	1.006 (±0.022)	1.006 (±0.020)	1.010 (±0.019)	0.988 (±0.033)	1.002 (±0.046)	1.016 (±0.028)	0.982 (±0.030)
	Chinese	Accuracy	0.547 (±0.026)	0.527 (±0.055)	0.536 (±0.023)	0.593 (±0.032)	0.680 (±0.034)	0.594 (±0.037)	0.429 (±0.033)
		Unbiasedness	1.070 (±0.075)	1.058 (±0.023)	1.067 (±0.068)	0.943 (±0.066)	0.904 (±0.053)	0.987 (±0.084)	1.200 (±0.114)

**Table 5 animals-13-00636-t005:** Accuracy of the younger individuals in the cross population prediction.

Reference	Validation	Milk Yield	Fat Yield	Protein Yield	Body Conformation	Feet and Legs	Mammary System Conformation	Somatic Cell Score
Chinese	Nordic	0.302	0.280	0.258	0.101	0.088	0.028	0.132
Nordic	Chinese	0.354	0.323	0.303	0.198	0.212	0.142	0.194

**Table 6 animals-13-00636-t006:** Accuracy of the cross-population prediction in the 10 replicates of the five-fold cross-validation.

Reference	Validation	Milk Yield	Fat Yield	Protein Yield	Body Conformation	Feet and Legs	Mammary System Conformation	Somatic Cell Score
Chinese	Nordic	0.378 (±0.030)	0.390 (±0.029)	0.364 (±0.025)	0.332 (±0.037)	0.068 (±0.035)	0.349 (±0.030)	0.142 (±0.033)
Nordic	Chinese	0.444 (±0.021)	0.468 (±0.028)	0.500 (±0.023)	0.323 (±0.035)	0.122 (±0.040)	0.303 (±0.033)	0.216 (±0.036)

## Data Availability

Not applicable.

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
