# Peer review of "Improving Genomic Prediction Accuracy in the Chinese Holstein Population by Combining with the Nordic Holstein Reference Population"

_animals, 2023, doi:10.3390/ani13040636_

Round 1

Reviewer 1 Report

Comments to the manuscript “Improving genomic prediction accuracy in Chinese Holsteins population by combining Nordic Holstein reference population” - animals-2072349

Mayor comments

In general, even the manuscript was well written, the topic is not new. In the introduction, author’s mention that LD concordance measured as correlation of 0.97, this argument probably guide the possible results of this study.

The definition of the references populations was not described properly. It is just mentions as cut-off date in the table 1.

I would suggest to the authors try some other research arguments to improve this study for example: include also levels of reliability. Probably, a cross validation could help too?

Please provide more details of the distributions of important details of the references and training populations in all scenarios.

Please provide enough arguments to continuing the revision in order to bring proper scientific contributions to take into account the manuscript.

Mino comments.

I will provide when authors return the improved version.

Reviewer 2 Report

Data used: authors have to explain their dataset and which animals were genotyped. Are these cows, bulls or both.

The datasets used here are rather historic and I wonder that not newer data sets were available. The outcomes of this study may be not very actual and not of actual worth. Indeed, nowadays datasets with genotyped Holsteins are rather big and may contain more than 500,000 animals.

Variances and covarinaces of traits analysed should be given.

Replicates on validation are missing. Authors do not present a 5-fold-cross-validation.

Another issue may influence the outcomes of this study. How far distant were the generations in the validation and reference populations. In case of closer related animals accuracies may be overestimated.

Reviewer 3 Report

The manuscript presents an interesting analysis to improve the prediction of breeding values of candidate animals using two different reference populations. This subject, which is important, has been previously studied using other populations. No novel methods or algorithms were evaluated. Thus, the subject essentially does not present enough novelty; however, I understood that it is important for Chinese Holstein Breeding Programs. The method is suitable and the result and discussion sections are easy-to-read and well-structured. I recommend accepting this manuscript after modifying some typos and clarifying some topics.

You can find the comments, modifications and suggestions attached here.  

Round 2

Reviewer 1 Report

One mMinor comments:

Please review

Table 1. Somatic cell score (kilo/ml).

Please try to review all the manuscripts once again in order to avoid misspelling word.

Author Response

Thank you. The authors have worked with a professional editing company to improve the quality of the presentation.

Reviewer 2 Report

The authors have replied to the questions and comments and the mansucript accordingly amended and improved.

Author Response

Thank you.